# Pathogen Detection via Impedance Spectroscopy-Based Biosensor

**DOI:** 10.3390/s24030856

**Published:** 2024-01-28

**Authors:** Tharun Reddy Kandukuri, Ioannis Prattis, Pelumi Oluwasanya, Luigi G. Occhipinti

**Affiliations:** Electrical Engineering Division, Department of Engineering, University of Cambridge, Cambridge CB3 0FA, UK; trk25@cam.ac.uk (T.R.K.); ip355@cam.ac.uk (I.P.)

**Keywords:** pathogen detection, ambient air, capacitive biosensors, bio-functionalized hydrogel, interdigitated electrodes (IDE), Influenza A

## Abstract

This paper presents the development of a miniaturized sensor device for selective detection of pathogens, specifically Influenza A Influenza virus, as an enveloped virus is relatively vulnerable to damaging environmental impacts. In consideration of environmental factors such as humidity and temperature, this particular pathogen proves to be an ideal choice for our study. It falls into the category of pathogens that pose greater challenges due to their susceptibility. An impedance biosensor was integrated into an existing platform and effectively separated and detected high concentrations of airborne pathogens. Bio-functionalized hydrogel-based detectors were utilized to analyze virus-containing particles. The sensor device demonstrated high sensitivity and specificity when exposed to varying concentrations of Influenza A virus ranging from 0.5 to 50 μg/mL. The sensitivity of the device for a 0.5 μg/mL analyte concentration was measured to be 695 Ω· mL/μg. Integration of this pathogen detector into a compact-design air quality monitoring device could foster the advancement of personal exposure monitoring applications. The proposed sensor device offers a promising approach for real-time pathogen detection in complex environmental settings.

## 1. Introduction

Air pollution, a significant environmental concern, impacts human health globally, causing an estimated 800,000 deaths annually in Europe and an alarming 8.8 million worldwide [1]. This issue is exacerbated by exposure to particulate matter (PM), identified as a major hazard. Moreover, the COVID-19 pandemic highlighted the critical role of airborne pathogens in disease transmission, underscoring the urgent need for effective detection mechanisms in ambient air [2].

This work addressed this challenge by developing a miniaturized sensor device for detecting pathogens. Influenza viruses, belonging to the Orthomyxoviridae family, are enveloped and feature a genome with segmented negative-sense single-strand RNA segments. Despite their susceptibility to environmental stressors, these viruses can endure for several hours under optimal conditions and, in colder water (below 20 °C), remain viable for several months. Sensitivity to lipid solvents, detergents, heat, and low pH, varying with the virus type, underscores their vulnerability. Notably, Influenza A viruses with uncleaved HA exhibit greater stability, maintaining infectivity at pH values lower than 4.5, in contrast to viruses with cleaved HA, which lose infectivity at a pH lower than 5 [3]. Knowing that PM could be a carrier for airborne pathogens, the integration of a bespoke biosensor with a particulate matter detection platform, combines pathogen detection with air quality monitoring [4,5] in a novel way and provides a more comprehensive understanding of air quality and health risks.

This work advanced an existing technology developed within our research group, which employs thermophoresis and capacitance measurement for particulate matter separation and detection [6]. The technology can differentiate smaller particles (PM2.5) from larger ones (PM10), enabling precise detection in various sections of the device [7]. This enhancement enables the particulate matter sensor to capture and identify airborne pathogens, potentially transforming air quality monitoring systems [8].

Through this integration, the proposed device technology provides vital insights into the spread of airborne diseases using impedance spectroscopy. Impedance biosensors, owing to their entirely electrical nature, stand out for their simplicity compared to other methods [9]. They do not incorporate optical or acoustic components, providing notable benefits for portable applications [10]. It offers a promising avenue for early detection and monitoring, contributing significantly to public health and safety. Electrochemical biosensors integrate an analyte-reception mechanism with an electrochemical transducer. The interaction between the targeted analyte and the transducer results in the generation of an electrochemical signal, manifested in current, potential, resistance, or impedance formats [11]. This innovative approach aligns with global efforts to combat air pollution and its associated health hazards, marking a pivotal advancement in environmental health research and technology.

## 2. Materials and Methods

### 2.1. Materials

The study utilized various materials and reagents, including PEGDA (Poly(ethylene glycol) diacrylate) obtained from Sigma Aldrich (Saint Louis, MO, USA), (2-Hydroxy-$4^{\prime }$-(2-hydroxyethoxy)-2-methyl-propiophenone) as the photo-initiator(PI), and PBS (phosphate-buffered saline) at 7.4 pH also obtained from Sigma Aldrich. Additionally, Recombinant Avian Influenza A (H5N8) Hemagglutinin protein (ab217663) at a 1mg/mL concentration and Anti-Avian Influenza A Hemagglutinin antibody (ab21297) at a 1mg/mL concentration were sourced from Abcam for experimental purposes.

### 2.2. Design

#### 2.2.1. IDE Fabrication

The IDE electrodes (Figure 1) were fabricated using E-beam photolithography on glass substrates to create a prototype pathogen detector. This fabrication method was chosen for its precision and ability to produce high-quality electrodes. The electrode configuration was optimized by enhancing gold adhesion to glass using a titanium layer and by employing oxygen plasma techniques for surface cleaning and improvement. The devices were printed on a microscopic slide measuring 75 by 26 mm.

#### 2.2.2. Hydrogel

To prevent damage to the biomaterial and ensure consistent results, a hydrogel was introduced on the IDE electrodes. The hydrogel serves as a virus capturing medium, retaining the virus within the sensor’s active area during measurements and providing measurement stability to the transducer. Initial experiments revealed that depositing antibodies without hydrogel, especially in small quantities at room temperatures, led to the rapid evaporation of the biomaterial from the electrode. The adoption of a hydrogel membrane over the sensor active area provides a semi-wet three-dimensional ideal environment for molecular-level biological interaction [12].

The hydrogel was formulated using a combination of PEGDA and PI, with specific attention paid to its optimization. A volume of 50 mL of hydrogel was prepared by mixing 20 g of PEGDA purchased by Sigma Aldrich, with 0.5 g of a photoinitiator—namely, (2-Hydroxy-$4^{\prime }$-(2-hydroxyethoxy)-2-methyl-propiophenone) sourced from Sigma Aldrich—was poured in a beaker. The beaker was then filled with DI water up to the 50 mL mark. The resulting solution was then sonicated for 10 min to improve its homogeneity. Subsequently, the homogenous solution was deposited onto the electrodes using a calibrated pipette, and the electrodes exposed to UV light for 9 min to initiate the crosslinking of the hydrogel promoted by the photoinitiator. This optimization process included ensuring that the hydrogel would not evaporate at room temperature, maintaining its viscosity to stay in droplet form on the electrodes, and achieving proper crosslinking through UV exposure. After depositing the hydrogel onto the IDEs using a pipette, it was subjected to UV light for gelation, a step that was crucial for ensuring the stability and functionality of the hydrogel in the pathogen detection application.

#### 2.2.3. Antibody and Analyte Preparation

First, specific antibodies of Influenza A were prepared at concentrations of 100 μg/mL and 50 μg/mL by diluting a 1 mg/mL antibody (AB) solution in phosphate-buffered saline (PBS). This preparation involved precise calculations to determine the correct amounts of PBS to add to the AB and the analyte to achieve the required concentrations. The entire process was conducted in a clean room environment to ensure the absence of contaminants that could interfere with the antibody’s activity. Following preparation, the antibody solutions were stored in a refrigerator at 4 degrees Celsius, as recommended by the manufacturer in the datasheet. This process was critical to maintain the stability and functionality of the antibodies for detecting the target pathogen.

To generate different diluted concentrations of the Influenza A analyte, appropriate dilutions were made using PBS. The concentrations used for testing and calibration included 0.5, 1, 5, 10, and 50 μg/mL. This range of concentrations allowed for a comprehensive evaluation of the sensor’s performance across different analyte concentrations. All tests and validation experiments reported in this manuscript were carried out by using a calibrated pipette from DLAB for introducing a fixed volume of 0.1 μL of analyte solution at different concentrations onto the sensor’s active area.

### 2.3. Laboratory Setup

The custom connector setup, depicted in Figure 2, ensured a secure and reliable connection during prototype testing. Spring-loaded connectors on the IDE devices minimized noise interference, maintaining proper contact with the potentiostat (EmStat4S, PalmSens, Houten, The Netherlands). Screw terminal blocks securely connected the IDE devices to the potentiostat, enabling accurate measurements and routed to a potentiostat connector to ensure a continuous and comprehensive characterization throughout.

### 2.4. Impedance Measurements

To assess the performance of the prototype pathogen detection device, impedance measurements were conducted using the potentiostat under different experimental conditions, as reported in Section 3. For accurate impedance response measurement, two-electrode reference electrode and counter electrode probes were connected to create a two-electrode measurement setup.

Central to the investigation were impedance measurements obtained from voltage sweeps at an operational frequency of 1000 Hz. These measurements played a crucial role in providing insights into the device’s responsiveness to varying concentrations of the Influenza A analyte. By analyzing these impedance variations at different analyte concentrations, we were able to effectively detect and quantify the presence of Influenza A. This approach not only enabled us to assess the device’s detection capabilities but also offered a detailed evaluation of its performance across various pathogen concentrations.

The subsequent subsections of our study focus on two primary types of data representations. Firstly, histogram graphs display the average impedance values corresponding to different concentrations of Influenza A, in the presence of fixed antibody concentrations. These graphs are augmented with linear regression analysis, offering a quantitative perspective on the relationship between impedance and analyte concentration. Secondly, we delve into sensitivity plots that illustrate the device’s sensitivity to different levels of pathogens. These plots are instrumental in quantifying the device’s response, as they examine the changes in impedance relative to the changes in analyte concentration, thereby providing a comprehensive understanding of the device’s capabilities in pathogen detection.

## 3. Results

This section introduces the results obtained from the comprehensive experiments with the interdigitated electrode (IDE) device, designed for pathogen detection. Our experimental framework encompassed a series of frequency sweeps, voltage sweeps, and time scan measurements, each tailored to unravel the device’s detection capabilities.

### 3.1. Frequency Sweep and Operational Frequency

To determine the optimal operational frequency for impedance measurements, several factors were considered, including sensitivity and response time. It was observed that higher frequency ranges allowed for faster signal acquisition but resulted in lower impedance values, as shown in Figure 3. Conversely, lower frequencies offered better sensitivity in terms of impedance values but required more time to acquire the sample. After careful evaluation, an operational frequency of 1000 Hz was chosen, as it fell below the 3 dB cutoff point, providing an optimal balance between sensitivity and response time for impedance measurements.

### 3.2. Voltage Sweep Measurements

In the series of experiments, voltage sweep measurements were carried out at an operational frequency of 1000 Hz. The interdigitated electrode (IDE) devices were evaluated under varying Influenza A analyte concentrations, juxtaposed with fixed antibody (AB) concentrations of 100μg/mL and 50μg/mL. The tested analyte concentrations included 0.5μg/mL, 1μg/mL, 5μg/mL, 10μg/mL, and 50μg/mL.

For the 100μg/mL AB concentration, the observed average impedance values were 1821Ω, 1473Ω, 1292Ω, 1258Ω, 1194Ω, and 1160Ω corresponding to control, 0.5μg/mL, 1μg/mL, 5μg/mL, 10μg/mL, and 50μg/mL analyte concentrations, respectively. For the 50μg/mL AB concentration, the impedance values were 1175Ω, 1101Ω, 1055Ω, 984Ω, 918Ω, and 866Ω. These values are graphically represented in Figure 4 and Figure 5, where average impedance values are plotted against analyte concentrations.

The linear regression analysis, illustrated by red dashed lines, quantitatively defines the relationship between impedance and analyte concentration. The slopes of these regression curves are indicative of the device’s responsiveness to varying analyte concentrations. As shown in Figure 4 and Figure 5, the relation between impedance and concentration is not linear. We used linear regression in the plot to highlight the presence of a decreasing trend. The percentage decrease in impedance from the control to the highest analyte concentration is notable: 36.3% for the 100μg/mL AB concentration and 26.3% for the 50μg/mL AB concentration.

These percentages highlight the higher responsiveness of the devices with higher concentration of antibodies which, up to a certain threshold, is attributable to a saturation of their binding sites. Beyond a certain concentration threshold, additional analyte molecules find no available binding sites, leading to a leveling off in impedance change. This saturation point is not only indicative of the device’s maximum detection limit but also sheds light on the efficiency of the antibody-analyte binding mechanism.

The methodical calculation of each average impedance value was a critical part of our analysis. This entailed conducting extensive voltage sweeps across various states of the device: the initial baseline state following hydrogel application, subsequent states post-antibody (AB) deposition using a pipette, and various stages after introducing different concentrations of analytes. By meticulously averaging the readings from these sweeps, we were able to construct detailed histogram graphs. These graphs not only visually represent the impedance responses under varying experimental scenarios but also incorporate linear regression analysis and sensitivity plots, enhancing our understanding of the device’s detection capabilities.

These sensitivity plots are particularly noteworthy, as they precisely illustrate the device’s response to specific analyte concentrations while maintaining a constant AB concentration. Such detailed sensitivity analysis, which we will delve into in the subsequent sections, is crucial for comprehending the full range and potential of the device in pathogen detection. The ability to accurately discern subtle changes in pathogen concentration has significant implications for the practical deployment of this technology in various health and environmental monitoring applications, potentially revolutionizing our approach to pathogen detection and air quality assessment.

### 3.3. Time Scan Measurements

Time scan measurements played a crucial role in our study, enabling us to observe the dynamic changes in impedance as the Influenza A analyte concentrations varied over time. Initially, the experiments commenced with a set concentration of 50 μg/mL Influenza A AB. Subsequently, there was a systematic introduction of increasing concentrations of the analyte at designated time intervals. This approach allowed for a comprehensive analysis of the impedance response over time under changing analyte conditions.

Figure 6 effectively captures these time scan measurements, showcasing the progression of impedance values at specific time points that align with the varying concentrations of the Influenza A analyte. The data reported in this figure and in Table 1 are instrumental in demonstrating how the device responds over time to changes in the analyte concentration, thereby highlighting the device’s temporal sensitivity to the Influenza A analyte.

Remarkably, the biosensor exhibited a high level of stability across diverse concentrations and states. The fluctuations in impedance readings averaged around a mere 1 % deviation from the mean values. This consistency is a testament to the sensor’s reliability in providing stable and accurate readings, a vital attribute for precise diagnostics and effective monitoring in real-world applications.

At a state where the AB concentration was maintained at 50 μg/mL, the maximum deviation recorded from the mean impedance value was a mere 1.2448%. The differential between the maximum fluctuation and the mean impedance stood at 56.357 Ohms, highlighting the sensor’s remarkable precision and nearly 99% stability of its readings. Such precision is exemplary in the field of biosensing, demonstrating the sensor’s capability to deliver accurate results with minimal variability.

The sensor’s performance was consistent even with varying concentrations of Influenza A, ranging from a low of 0.5 μg/mL to as high as 50 μg/mL. At 0.5 μg/mL, the fluctuation was a minimal 0.59146% from the mean. Impressively, even with tenfold (5 μg/mL) and hundredfold (50 μg/mL) increases in the analyte concentration, the fluctuations remained low at 0.42141% and 0.4339%, respectively. These small fluctuations are indicative of the sensor’s robust performance, capable of maintaining a high level of stability across a wide range of analyte concentrations. This consistency is crucial for a reliable biosensor, ensuring that it can adapt to detect varying levels of pathogens in different environmental conditions.

The quick stabilization time further enhances the sensor’s practicality. The sensor’s readings stabilize within a minute, as demonstrated by the consistent measurements observed in time brackets ranging from 30–70 s, 130–170 s, and continuing in similar intervals thereafter. This rapid return to stability after each reading signifies that the sensor is not only stable but also responsive, making it ideal for successive testing scenarios where time efficiency is paramount.

The combination of great stability and swift stabilization times positions this biosensor as a highly reliable tool for ongoing viral monitoring, even when dealing with the inherently dynamic biological samples. Such stability is vital for real-world applications where consistent and rapid readings can significantly impact the outcomes of disease detection and management.

## 4. Discussion

### 4.1. Sensitivity Analysis

The sensitivity of the biosensor was rigorously tested by quantifying its electrical response in Ω·mL/μg, with a particular focus on Influenza A virus to emulate real-world pathogen presence. The calibration phase involved a baseline impedance measurement with a control solution, which was crucial for comparing the impact of subsequent analyte introductions. In a series of meticulous trials, the sensor was exposed to Influenza A virus in concentrations carefully escalated from 0.5 to 50 μg/mL. These test concentrations were thoughtfully chosen to reflect a range from the sensitivity threshold of the sensor, starting at 10 particles/ml, equivalent to 0.5 μg/mL, to the upper detection limit of 100,000 particles/mL, which converts to 8 × 10 μg/mL. This span encompasses the typical viral particle concentration found in human coughs and the estimated weight of individual virus particles, drawing on insights from previous studies [13,14,15].

Examining the sensitivity plots for the 50 μg/mL and 100 μg/mL AB concentrations in Figure 5 and Figure 6, we can derive quantitative insights into the biosensor’s performance.

In the 50 μg/mL sensitivity plot, the sensor exhibits a high sensitivity value of 148 Ω·mL/μg at the lowest analyte concentration, which drastically decreases as the concentration increases, reaching values such as 120 Ω·mL/μg, 38 Ω·mL/μg, 26 Ω·mL/μg, and stabilizing around 6 Ω·mL/μg at higher concentrations. This demonstrates a high degree of sensitivity at low concentrations, crucial for the detection of trace amounts of analytes. The steep initial slope of the linear regression curve indicates that the sensor’s impedance is highly responsive to changes in concentration at low levels, which is desirable in a sensitive detection system.

The sensitivity plot for the 100 μg/mL AB concentration begins with an even higher sensitivity of 695 Ω·mL/μg, suggesting that the biosensor has an excellent initial response when encountering low analyte levels. The sensitivity decreases as the concentration increases, marked by subsequent values of 529 Ω·mL/μg, 112 Ω·mL/μg, 63 Ω·mL/μg, and 13 Ω·mL/μg. Similar to the 50 μg/mL plot, there is a distinct drop-off in sensitivity as the concentration rises, indicative of a sensor with a high dynamic range in the lower concentration spectrum.

Both plots confirm that the biosensor’s impedance measurement is sensitive to the concentration of the viral analyte. The linear regression lines in these plots serve as quantitative tools for interpreting the sensor’s performance, with the slope representing the rate at which sensitivity changes with concentration. The plots effectively showcase the biosensor’s capability for quantitatively analyzing the impedance–analyte concentration relationship, which is essential for its application in sensitive diagnostic settings.

### 4.2. Comparison and Novelty

This subsection presents a comparative analysis of the findings of our work and research conducted by others in the field. The biosensor developed in this study exhibits remarkable sensitivity and response time when benchmarked against state-of-the-art sensors as outlined in Table 2. For instance, the sensor’s ability to detect Influenza A at a concentration of 500 ng ${\mathrm{mL}}^{-1}$ within just 1 min is a significant advancement over the sensor by Nidzworski et al. [16], which requires 30 min to detect a minimum of 20 pg ${\mathrm{mL}}^{-1}$ of Influenza. This improvement in detection speed is critical for rapid diagnostics and immediate patient care decisions.

Furthermore, compared to the biosensors cited for JEV antigen and Dengue Virus detection, which have a minimum detectable limit of 0.75 μg ${\mathrm{mL}}^{-1}$ and 1 pfu ${\mathrm{mL}}^{-1}$, respectively, the current study’s sensor demonstrates an equally good detection limit, enhancing its utility in early stage disease detection where viral loads are often minimal. This novel range and sensitivity underscore the biosensor’s potential as a versatile tool in the early detection and management of viral infections.

### 4.3. Selectivity of AB ab21297 and Its Impact on Impedance Variation

The selectivity of the antibody (AB) ab21297 is a pivotal aspect in the functionality of our biosensor, as demonstrated by the impedance measurements illustrated in Figure 7. This specific AB is meticulously engineered to selectively bind with the influenza virus, a feature that is paramount in biosensing applications. The sensor specificity relies on the use of the specific antibody for influenza A, which would not bind other viruses or pathogens [25]. This high level of specificity ensures that ab21297 does not inadvertently interact with other substances, thereby significantly reducing the likelihood of false positive results and enhancing the overall reliability of the detection process.

The Anti-Avian Influenza A Hemagglutinin antibody is a synthetic peptide corresponding to a segment near the N terminus of the Hemagglutinin protein. The antibody specifically recognizes the cleaved subunit of Hemagglutinin but not the full-length H5. Hemagglutinin (HA) is located on the virus’s surface and plays a key role in virus entry into the host cell by binding to receptors.

In electrochemical impedance spectroscopy sensors, a difference in the electrical signal is created due to the kinetic binding of antibodies and its antigens at the sensors surface. As a result, electron transfer/charge transfer resistance is produced, representing the number of bound molecules [26].

Our experimental setup rigorously tested this selectivity, and the results were evident in the distinct impedance variations corresponding to different phases of the biosensor’s operation. Initially, with the AB ab21297 deposited using pipette onto the device, a baseline impedance was established, setting a reference point for further measurements. Subsequent pipette deposition of DI water onto the AB-coated device led to an observable increase in impedance. This increase is primarily attributed to the lack of interaction between the antibodies and DI water, as ab21297 is designed to be unreactive to substances other than its intended target analyte.

The experiment, when the intended analyte, was introduced to the AB-coated surface. As shown in Figure 7, the specific bonding of ab21297 with the Influenza virus resulted in a notable decrease in impedance, a phenomenon that is directly linked to an increase in conductivity due to the AB–analyte interaction. This response not only reaffirmed the AB’s selectivity but also highlighted its efficacy in modifying the biosensor’s electrical characteristics upon the presence of the target virus.

The selective binding mechanism and its consequent effect on impedance are crucial indicators of the biosensor’s efficiency. The AB’s capability to specifically bind with the influenza virus and influence the electrical properties of the sensor is a testament to the critical role of selecting highly specific antibodies in the development of biosensors. This specificity transcends being a mere feature; it is an essential requirement in diagnostic applications where precision and accuracy are of utmost importance. The impedance variations, as depicted in the histogram, provide a quantitative measure of this selective interaction, thereby underlining the indispensable role of AB ab21297 in the biosensor’s design and operational framework. This specificity is vital in distinguishing between the target virus and other potentially present entities, ensuring the biosensor’s precision and applicability in real-world scenarios.

Further, the impedance changes observed in our experiments align with theoretical predictions and previous studies, confirming the AB’s affinity for the influenza virus and its negligible interaction with other particles or substances. This adds another layer of validation to our biosensor’s design, indicating that it can reliably differentiate between the presence and absence of the target pathogen. This specificity is particularly important in complex environmental samples where various substances coexist, highlighting the biosensor’s potential for application in diverse settings.

In summary, the role of AB ab21297 in our biosensor is multi-faceted. It not only provides the necessary selectivity for accurate pathogen detection but also paves the way for innovative enhancements in biosensor technology. The impedance changes observed in our experiments are not just measurements; they are a testament to the potential of highly specific ABs in revolutionizing biosensing and diagnostic methodologies.

## 5. Conclusions and Future Work

### 5.1. Conclusions

The innovative IDE biosensor described in this paper presents a significant advancement in the field of pathogen detection. The experiments have conclusively demonstrated the sensor’s capabilities, particularly in detecting and quantifying Influenza A analytes with remarkable precision. The IDE sensor operates optimally at a frequency of 1000 Hz, a parameter established through frequency sweep measurements to ensure a balance between sensitivity and expediency.

Further reinforcing its efficacy, voltage sweep and time scan assessments have verified the sensor’s ability to discern minute variations in analyte concentrations, a testament to its fine-tuned detection acumen. The stability and responsiveness of the sensor, as evidenced by minimal fluctuations in readings and rapid stabilization times, are indicative of its reliability and practical utility.

The impedance measurements, complemented by thorough linear regression analysis, have laid the foundation for a detailed understanding of the sensor’s interaction with various analyte concentrations. The sensitivity plots generated have not only delineated the sensor’s detection range but have also served as a calibration tool, crucial for the sensor’s practical deployment in diagnostic settings.

The device’s primary limitation is that it cannot be reused without thorough cleaning and reapplication of hydrogel and antibodies. However, this constraint does not impede its ability to perform continuous time measurements, provided there are sufficient binding sites available for antigen attachment.

In essence, this IDE device is not only a harbinger of promise for pathogen detection but also an embodiment of innovation in biosensing technology. With its ability to provide rapid, stable, and precise measurements, it holds the potential to revolutionize early diagnosis and monitoring of infectious diseases.

### 5.2. Future Steps

Based on successful outcomes, our next steps involve the further development and integration of the IDE device. The device will be miniaturized for improved portability and combined with particulate matter detection technology for comprehensive sensing capabilities.

We aim to embed the IDE device into a portable frontend electronics system, enabling pathogen detection in remote locations. User-friendly interface and data analysis algorithms will be developed to facilitate result interpretation.

Future research will focus on expanding pathogen detection capabilities, optimizing sensitivity and specificity for different analytes. Ongoing refinement and validation will ensure reliability in diverse environmental conditions.

By advancing the IDE device and integrating relevant technologies, we contribute to proactive monitoring and timely interventions for disease control.

In summary, our research demonstrates the potential of the IDE device as a valuable tool for pathogen detection. Successful outcomes pave the way for further advancements, making rapid and reliable pathogen detection accessible for public health and safety.

## Figures and Tables

**Figure 1 sensors-24-00856-f001:**
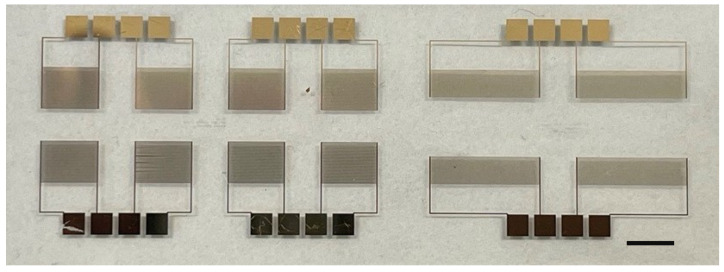
IDEs on a glass substrate made via E-beam lithography. Scale bar = 5 mm.

**Figure 2 sensors-24-00856-f002:**
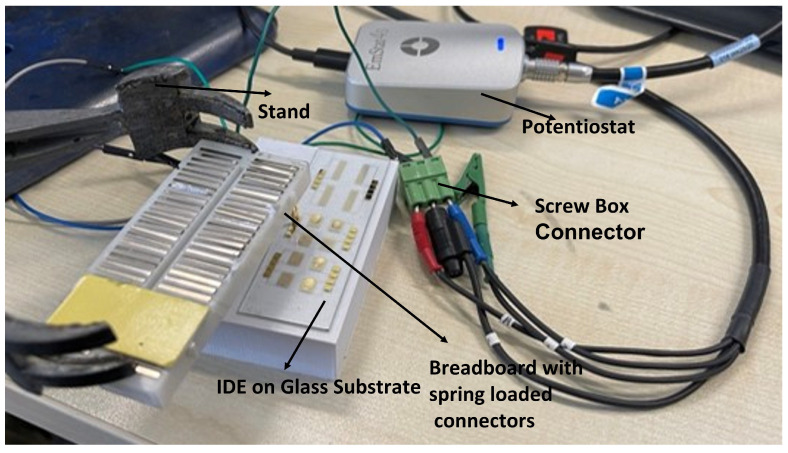
Testing setup showing a custom connector incorporating spring-loaded pins, a stand, and a screw terminal box connector to establish a secure and reliable connection between components.

**Figure 3 sensors-24-00856-f003:**
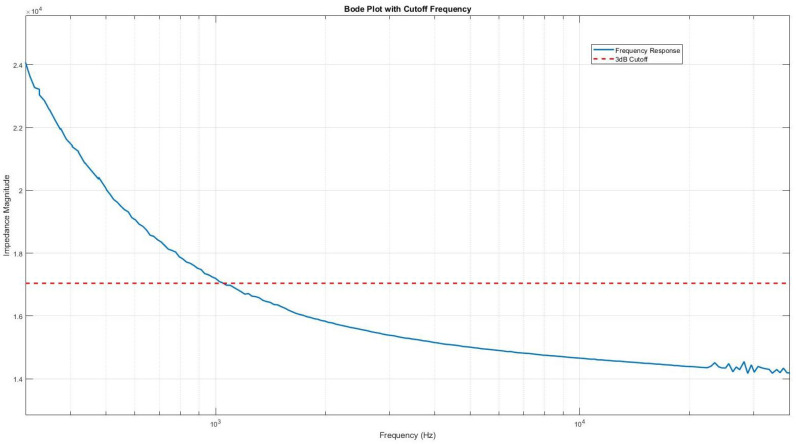
Bode diagram (magnitude) of the device after hydrogel has been deposited using pipette (blue line) and 3dB cutoff frequency (red dotted line).

**Figure 4 sensors-24-00856-f004:**
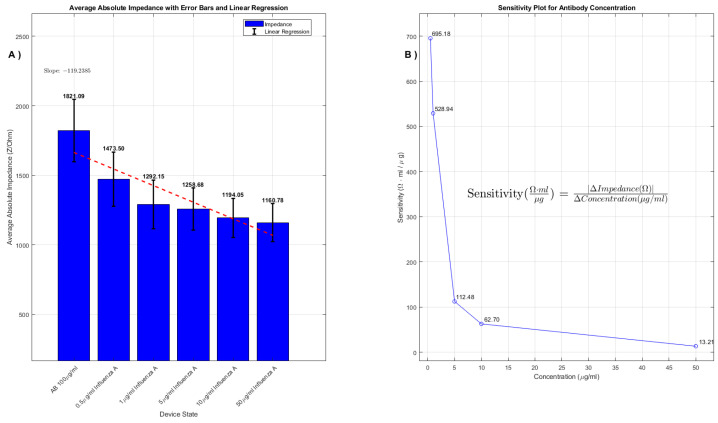
(**A**) The average impedance values collected with voltage sweep during voltage sweep (−1 V to 1 V) for AB 100μg/mL. The bar plot demonstrates the average absolute impedance for different concentrations of Influenza A, while the red dashed line indicates the linear regression curve. The slope of this line reveals the rate of change in impedance in relation to Influenza A concentration. (**B**) Sensitivity plot of impedance change with concentration of analyte: the sensitivity of the device to different concentrations of Influenza A at a constant AB concentration of 100μg/mL. The sensitivity formula is articulated as sensitivity ($\frac{\Omega \cdot \mathrm{mL}}{\mu g}$) = $\frac{\left|\Delta Impedance(\Omega )\right|}{\Delta Concentration(\mu g/\mathrm{mL})}$.

**Figure 5 sensors-24-00856-f005:**
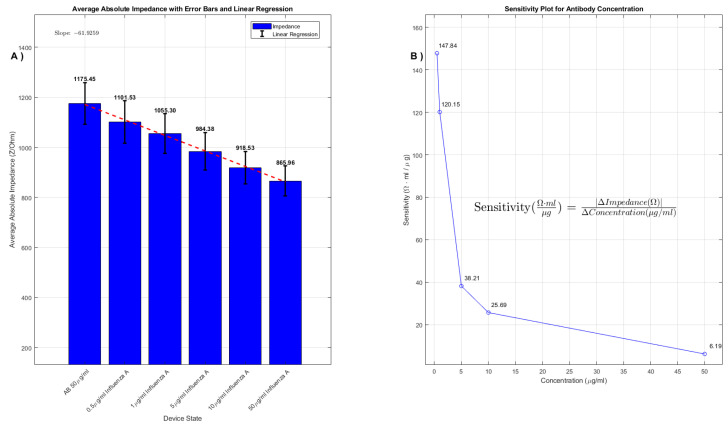
(**A**) The average impedance values obtained from a voltage sweep during voltage sweep (−1 V to 1 V) for AB 50μg/mL. The bar plot illustrates the average absolute impedance at varying concentrations of Influenza A, with the red dashed line representing the linear regression curve. This slope indicates the change in impedance as a function of Influenza A concentration. (**B**) The sensitivity plot of impedance change versus concentration of analyte, demonstrating the device’s sensitivity to changes in Influenza A concentration at a fixed AB concentration of 50μg/mL. The equation for sensitivity is given by Sensitivity ($\frac{\Omega \cdot \mathrm{mL}}{\mu g}$) = $\frac{\left|\Delta Impedance(\Omega )\right|}{\Delta Concentration(\mu g/\mathrm{mL})}$, indicating a quantifiable relationship between impedance change and analyte concentration.

**Figure 6 sensors-24-00856-f006:**
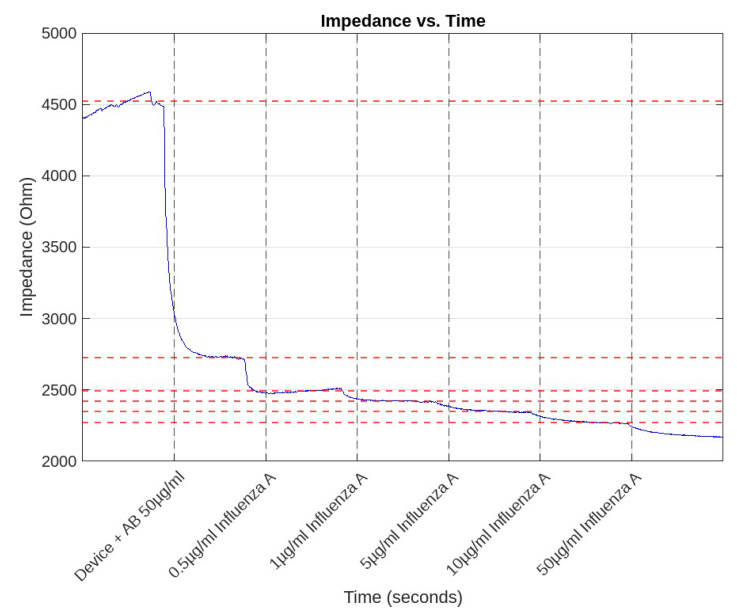
Time scan of impedance change with increasing concentrations of analyte (AB concentration: 50 μg/mL). The red dotted lines refer to the mean impedance values in the first column of Table 1.

**Figure 7 sensors-24-00856-f007:**
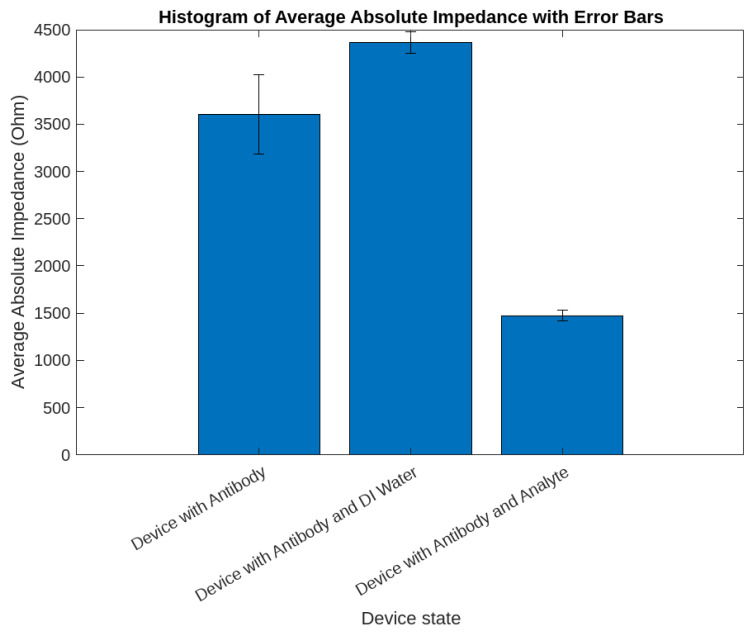
Comparative histogram analysis of average absolute impedance with error bars for three distinct device states: (**left**) Baseline impedance characteristics with antibodies deposited using pipette. (**middle**) Impedance changes after the deposition of DI water using pipette on the AB-coated surface. (**right**) Device response upon the deposition of the analyte using pipette on the AB-coated surface, indicating the final operational state. Each histogram bar is complemented by error bars indicating the standard deviation from multiple readings, underscoring the measurements’ reproducibility and reliability. (AB concentration: 100 μg/mL).

**Table 1 sensors-24-00856-t001:** Statistical information on time scan graph.

Mean Impedance	Std Dev	Avg Percent Fluct	Max Fluctuation%	Diff MaxFluct Mean	Device State
4527.3	31.277	0.60411	1.2448	56.357	AB 50 μg/mL
2730.5	4.9103	0.12299	0.59146	10.939	0.5 μg/mL Influenza A
2493.8	5.1328	0.15799	0.551	11.662	1 μg/mL Influenza A
2421.3	2.9141	0.080546	0.42141	4.0774	5 μg/mL Influenza A
2350.3	4.0663	0.14494	0.36798	7.6768	10 μg/mL Influenza A
2274.9	4.3498	0.16243	0.4339	9.8708	50 μg/mL Influenza A

Note: table contains statistical data relating the analysis of time scan. Fluct stands for fluctuation.

**Table 2 sensors-24-00856-t002:** State-of-the-art electrochemical impedance spectroscopy-based biosensors.

Sample	Sensitivity/Time	Range
JEV Antigen	0.75 μg ${\mathrm{mL}}^{-1}$/20 min	1–10.0 μg ${\mathrm{mL}}^{-1}$ [17]
Dengue Virus	1 pfu ${\mathrm{mL}}^{-1}$/1 h 30 min	1 to 900 pfu ${\mathrm{mL}}^{-1}$ [18]
VSV (Virus)	$10^{4}$ pfu ${\mathrm{mL}}^{-1}$/–	$5\times 10^{4}$ to $5\times 10^{6}$ pfu ${\mathrm{mL}}^{-1}$ [19]
Influenza	20 pg ${\mathrm{mL}}^{-1}$/30 min	20–100 pg ${\mathrm{mL}}^{-1}$ [16]
Norovirus	1.7 copies ${\mathrm{mL}}^{-1}$/1.5 h	0 to $10^{5}$ copies ${\mathrm{mL}}^{-1}$ [20]
RABV	0.5 μg ${\mathrm{mL}}^{-1}$/1 h	0.1–4 μg ${\mathrm{mL}}^{-1}$ [21]
A/H7N1 Virus	5 μg ${\mathrm{mL}}^{-1}$/–	– [22]
Influenza A Virus (IAV)	8 ng ${\mathrm{mL}}^{-1}$/1 h	0 to 64 ng ${\mathrm{mL}}^{-1}$ [23]
Japanese Encephalitis Virus (PANI NWs)	<10 ng ${\mathrm{mL}}^{-1}$/10 min	– [24]
Influenza A	500 ng ${\mathrm{mL}}^{-1}$/1 min	500 ng ${\mathrm{mL}}^{-1}$ to 50 μg ${\mathrm{mL}}^{-1}$/This Paper

## Data Availability

The data presented in this study are available on request from the corresponding author.

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
