# Peer review of "Pathogen Detection via Impedance Spectroscopy-Based Biosensor"

_sensors, 2024, doi:10.3390/s24030856_

Round 1

Reviewer 1 Report

Comments and Suggestions for Authors

Comments on the Quality of English Language

Author Response

Authors response to Reviewer No. 1

We appreciate the valuable comments by this reviewer, which are addressed in the revised version of the manuscript as follows.

Comments

  1. The abstract appears to be concise, but it could benefit from additional points, particularly, about influenza A to provide a more comprehensive overview of the topic. Including key points and findings of developed biosensor would enhance the reader's understanding and engagement with the research.

Response: we thank this reviewer for the valuable comments. We have addressed this comment by adding the following sentences:

in the Abstract (line 2): “, specifically Influenza A. Influenza virus as an enveloped virus is relatively vulnerable to damaging environmental impacts. In consideration of environmental factors such as humidity and temperature, this particular pathogen proves to be an ideal choice for our study. It falls into the category of pathogens that pose greater challenges due to their susceptibility.”

and in the Introduction (starting at line 24 in the revised manuscript): “Influenza viruses, belonging to the Orthomyxoviridae family, are enveloped and feature a genome with segmented negative-sense single-strand RNA segments. Despite their susceptibility to environmental stressors, these viruses can endure for several hours under optimal conditions and, in colder water (below 20 °C), remain viable for several months. Sensitivity to lipid solvents, detergents, heat, and low pH, varying with the virus type, underscores their vulnerability. Notably, Influenza A viruses with uncleaved hemagglutinin (HA) exhibit greater stability, maintaining infectivity at a pH lower than 4.5, in contrast to viruses with cleaved HA, which lose infectivity at a pH lower than 5. [3]”

and added the following reference:

[3] Arbeitskreis Blut, Untergruppe «Bewertung Blutassoziierter Krankheitserreger». Influenza Virus. Transfus Med Hemother. 2009;36(1):32-39. doi: 10.1159/000197314. PMID: 21048819; PMCID: PMC2928832.

  1. It would be valuable to explore and compare the advantages of using impedance spectroscopy-based biosensors in greater detail. Highlighting the specific strengths and potential benefits of this method over alternative approaches would enhance the depth and relevance of the discussion, providing a more comprehensive understanding for readers.

Response: we are grateful to this reviewer for this comment. We have included the following text in the Introduction (line 43 of the revised manuscript):

“Impedance biosensors, owing to their entirely electrical nature, stand out for their simplicity compared to other methods [9]. They do not incorporate optical or acoustic components, providing notable benefits for portable applications [10].”

and added the following references:

[9] Radhakrishnan, Rajeswaran; Suni, Ian I.; Bever, Candace S.; Hammock, Bruce D. Impedance Biosensors: Applications to Sustainability and Remaining Technical Challenges. ACS Sustainable Chemistry & Engineering. 2014;2(7):1649-1655. doi: 10.1021/sc500106y.

[10] Laschi, Serena; Mascini, Marco. Planar electrochemical sensors for biomedical applications. Medical Engineering & Physics. 2006;28(10):934-943. ISSN 1350-4533. doi: 10.1016/j.medengphy.2006.05.006

  1. The caption for 'Table 2' on page 9 needs clarification or adjustment. It seems to convey information about the placement of tables but lacks specific details. Consider providing a more informative and concise description for better reader understanding.

Response: We thank the reviewer for spotting this mistake. We corrected the caption of table 9 in the revised manuscript as follows:

“State-of-the-art electrochemical impedance spectroscopy-based biosensors”

Consider revisiting the text to ensure thorough proofreading for English language and grammar. This will enhance the overall clarity and professionalism of the document.

Response: we revised the text and corrected the English language and grammar throughout the entire manuscript.

Abstract:

  • The sensor device demonstrated high sensitivity and specificity when exposed to varying concentrations of Influenza A virus, ranging from 0.5 to 50 mg/mL. Remove the unnecessary comma after "virus" for improved clarity.

Response: the comma has been removed.

  • The sensitivity of the device for 0.5 mg/mL analyte concentration was measured to be 810 W mL/mg. Add "a" before "0.5 mg/mL" for proper grammar.

Response: we added the missing “a” before “0.5 mg/mL”.

  • Integration of this pathogen detector into a compact design air quality monitoring device could foster the advancement of personal exposure monitoring applications. Consider adding a hyphen between "compact design" for consistency: "compact-design air quality monitoring device."

Response: we replaced “compact design” with “compact-design”.

Introduction:

  • Moreover, the COVID-19 pandemic highlighted the critical role of airborne pathogens p in disease transmission, underscoring the urgent need for effective detection mechanisms in ambient air[2]. Remove "p" after "pathogens" for clarity.

Response: corrected the typo.

  • Knowing that PM could be a carrier for airborne pathogens, the integration of a bespoke biosensor with a particulate matter detection platform, combines pathogen detection with air quality monitoring [3,4] in a novel. provides a more comprehensive understanding of air quality and health risks. Clarify the sentence for smoother flow.

Response: clarified the sentence by replacing “in a novel. provides” with “in a novel way and provides a”.

  • This work advanced an existing technology developed with in our research group, which employs thermophoresis and capacitance measurement for particulate matter separation and detection [5]. Change "with in" to "within" for proper grammar.

Response: typo corrected.

  1. Materials and Methods
  • Additionally, Recombinant Avian Influenza A (H5N8) Hemagglutinin protein (ab217663) at 1mg/ml concentration and Anti-Avian Influenza A Hemagglutinin antibody (ab21297) at 1mg/ml concentration were sourced from Abcam for experimental purposes. Add "a" before "1mg/ml" for proper grammar.

Response: added "a" before "1mg/ml"

  • The electrode configuration was optimized by enhancing gold adhesion to glass using a titanium layer and employing oxygen plasma techniques for surface cleaning and improvement. Add “by” before employing for proper grammar.

Response: added “by” before “employing”

Results

  • In this sentence, the phrase "3 dB cutoff" seems to be missing a verb or clarification: Original: "After careful evaluation, an operational frequency of 1000 Hz was chosen as it fell below the 3 dB cutoff, providing an optimal balance between sensitivity and response time for impedance measurements." Suggesting: "After careful evaluation, an operational frequency of 1000 Hz was chosen because it fell below the 3 dB cutoff point, providing an optimal balance between sensitivity and response time for impedance measurements."

Response: replaced “3dB cutoff” with “3dB cutoff point”

Reviewer 2 Report

Comments and Suggestions for Authors

In the paper titled "Pathogen detection via impedance spectroscopy-based biosensor," the authors present the development of a miniaturized, hydrogel-based impedance biosensor integrated into an existing platform for sensitive detection of high concentrations of airborne influenza pathogens. While this work shows some promise, the reviewer believes the paper barely meets the standards for publication as written. Offering the following constructive feedback could  strengthen the clarity and quality of the work:

  1. Please provide more details on how the analyte was introduced to the sensor. Was it added directly, or did you use any kind of delivery mechanism?
  2. Have you conducted any tests with analytes other than influenza? Reporting on the sensor's specificity with additional analytes would strengthen the study.
  3. The description of the hydrogel preparation lacks some key details needed to critically evaluate or replicate it. Specifically, reporting the pH of the PBS used and providing the full hydrogel recipe with measured component amounts would greatly improve this section.
  4. Consider adding a scale bar to Figure 1 to provide readers with a size reference.
  5. There are opportunities to reduce the word count substantially by removing repeated texts and excess descriptive language that does not directly support the key findings. For instance, the last sentence in Figure 7 and the concluding sentences in section 3.3 paragraphs would likely be removed or condensed without losing meaning.
Comments on the Quality of English Language

Overall the manuscript should be revised for English correction. 

Author Response

Authors response to Reviewer No. 2

Comments

Overall the manuscript should be revised for English correction.

Response: the manuscript has now undergone a full proofreading and English correction.

In the paper titled "Pathogen detection via impedance spectroscopy-based biosensor," the authors present the development of a miniaturized, hydrogel-based impedance biosensor integrated into an existing platform for sensitive detection of high concentrations of airborne influenza pathogens. While this work shows some promise, the reviewer believes the paper barely meets the standards for publication as written.

Response: We appreciate the feedback by this reviewer. We have revised thoroughly the manuscript and addressed all reviewers’ comments. We are confident the revised version of the manuscript is significantly improved both in its content and quality for publication.

Offering the following constructive feedback could strengthen the clarity and quality of the work:

  1. Please provide more details on how the analyte was introduced to the sensor. Was it added directly, or did you use any kind of delivery mechanism?

Response: we used a calibrated pipette with fixed volume of 0.1 µl. We have addressed this comment by adding this detail at the end of the sub-section 2.2.3 of the “materials and methods” section lines 107-110 as follows:

“All tests and validation experiments reported in this manuscript were carried out by using a calibrated pipette from DLAB for introducing a fixed volume of 0.1 µl of analyte solution at different concentrations onto the sensor’s active area.”

  1. Have you conducted any tests with analytes other than influenza? Reporting on the sensor's specificity with additional analytes would strengthen the study.

Response: we thank the reviewer for this question and comment. Indeed, we did not conduct tests with analytes other than influenza A, as the reported experiments were deemed sufficient to demonstrate the sensor concept and its viability. In fact, the sensor specificity relies on the use of the specific antibody for influenza A, which is designed to not bind other viruses or pathogens. This is corroborated by the characteristics of the antibody reported in its technical datasheet under the section “Specificity”: “anti-Hemagglutinin (NT) protein polyclonal antibody was raised against a synthetic peptide corresponding to 15 amino acids at the amino terminus of the Hemagglutinin protein (Genbank accession no. AAT76166). Efforts were made to use relatively conserved regions of the viral sequence as the antigen. The antibody only recognizes the cleaved subunit not the full-length H5”, whereby H5 stands for Hemagglutinin type 5. We have clarified this aspect in the revised version of the manuscript (lines 295-297) as follows:

“The sensor specificity relies on the use of the specific antibody for influenza A, which is designed to not bind other viruses or pathogens [17]”

And added the following reference:

[17] [abcam]. Anti-Avian Influenza A Hemagglutinin antibody ab21297. https://www.abcam.com/en-dk/products/primary-antibodies/avian-influenza-a-hemagglutinin-antibody-ab21297 (accessed 17 January 2024).

  1. The description of the hydrogel preparation lacks some key details needed to critically evaluate or replicate it. Specifically, reporting the pH of the PBS used and providing the full hydrogel recipe with measured component amounts would greatly improve this section.

Response: we thank the reviewer for this valuable suggestion. We have now included the details by adding the following text in sub section 2.2.2 (lines 81-88) of the revised manuscript:

“A volume of 50 ml of hydrogel was prepared by mixing 20 grams of PEGDA purchased by Sigma Aldrich with 0.5 grams of a Photoinitiator namely (2-Hydroxy-4'-(2-hydroxyethoxy)-2-methyl-propiophenone) sourced from Sigma Aldrich was poured in a beaker. The beaker was then filled with DI water up to the 50 ml mark. The resulting solution was then sonicated for 10 minutes to improve its homogeneity. Subsequently, the homogenous solution was deposited onto the electrodes using a calibrated pipette, and the electrodes exposed to UV light for 9 minutes to initiate the crosslinking of the hydrogel promoted by the photoinitiator.”

  1. Consider adding a scale bar to Figure 1 to provide readers with a size reference.

    Response: we added the scale-bar in Figure 1 as suggested. We have also added the following text in the revised version (lines 69-70):

“The devices were printed on a microscopic slide measuring 75 by 26 mm.”

  1. There are opportunities to reduce the word count substantially by removing repeated texts and excess descriptive language that does not directly support the key findings. For instance, the last sentence in Figure 7 and the concluding sentences in section 3.3 paragraphs would likely be removed or condensed without losing meaning.

Response: we thank the reviewer for this valuable comment and suggestion. We removed the repeated text in section 3.3 and last sentence in section 4.3.

Reviewer 3 Report

Comments and Suggestions for Authors

The manuscript, Pathogen detection via impedance spectroscopy-based biosensor, presents  the development of a miniaturized sensor device for selective detection of pathogens. The manuscript needs some major changes. The abstract and the introduction sections are very brief. Please add further information about the study design to the abstract section. The introduction section should be further expanded to address the background information; currently, the introduction is too brief and does not clearly provide enough information for the readers. The authors may benefit from the recent comprehensive review paper titled: Electrochemical Biosensors for Pathogen Detection: An Updated Review.

The discussion section should be eloborated with including more previous studies in the context to strengthen the discussion. The limitations of the study should be added. 

Comments on the Quality of English Language

minor errors

Author Response

Authors response to Reviewer No. 3

Comments

The manuscript, Pathogen detection via impedance spectroscopy-based biosensor, presents the development of a miniaturized sensor device for selective detection of pathogens. The manuscript needs some major changes.

Response: we appreciate the feedback by this reviewer. We have indeed implemented major changes by addressing all reviewers’ comments in the revised version of the manuscript.

  1. The abstract and the introduction sections are very brief. Please add further information about the study design to the abstract section.

Response: we have added the following text in the abstract (lines 2-5 of the revised manuscript):

“Influenza virus as an enveloped virus is relatively vulnerable to damaging environmental impacts. In consideration of environmental factors such as humidity and temperature, this particular pathogen proves to be an ideal choice for our study. It falls into the category of pathogens that pose greater challenges due to their susceptibility.”

  1. The introduction section should be further expanded to address the background information; currently, the introduction is too brief and does not clearly provide enough information for the readers. The authors may benefit from the recent comprehensive review paper titled: Electrochemical Biosensors for Pathogen Detection: An Updated Review.

Response: we have introduced the following text in the Introduction (line 47-51 of the revised manuscript):

“Electrochemical biosensors integrate an analyte-receptor binding mechanism with an electrochemical transducer. The interaction between the targeted analyte and the transducer results in the generation of an electrochemical signal, manifested in current, potential, resistance, or impedance formats.” [11]

and added the following reference:

[11] Banakar, M.; Hamidi, M.; Khurshid, Z.; Zafar, M.S.; Sapkota, J.; Azizian, R.; Rokaya, D. Electrochemical Biosensors for Pathogen Detection: An Updated Review. Biosensors. 2022;12, 927. doi: 10.3390/bios12110927.

  1. The discussion section should be elaborated with including more previous studies in the context to strengthen the discussion.

Response: we have added the following references to previous studies in the revised version of the manuscript:

[23] Hassen, Walid Mohamed; Duplan, Valérie; Frost, Eric; Dubowski, Jan J. Quantitation of influenza A virus in the presence of extraneous protein using electrochemical impedance spectroscopy. Electrochimica Acta. 2011;56(24):8325-8328. ISSN 0013-4686. doi: 10.1016/j.electacta.2011.07.009.

[24] Chu Van Tuan, Tran Quang Huy, Nguyen Van Hieu, Mai Anh Tuan, Tran Trung. Polyaniline Nanowires-Based Electrochemical Immunosensor for Label-Free Detection of Japanese Encephalitis Virus. Analytical Letters. 2013;46(8):1229-1240. DOI: 10.1080/00032719.2012.755688.

  1. The limitations of the study should be added.

Response: we have added the following text on the limitations of the study in the conclusions section of the revised manuscript (lines 364-367):

“The device's primary limitation is that it cannot be reused without thorough cleaning and reapplication of hydrogel and antibodies. However, this constraint does not impede its ability to perform continuous time measurements, provided there are sufficient binding sites available for antigen attachment”

Reviewer 4 Report

Comments and Suggestions for Authors

The authors designed an impedance sensor to detect influenza A virus antigen. The device is responsive to antigen concentration as low as 0.5 ug/mL. The idea is novel, but data interpretation in this paper needs improvement. The following comments need to be addressed before publication. 

1. The authors did linear regression analysis for "impedance vs antigen concentration" plots in Figure 4 and 5. However, the concentrations on the x axis are not in linear scale. Why did the author do linear regression analysis? It is not quantitative. Why the decreasing trends differ between 50 and 100 ug/mL antibody conditions? 

2. In line 140, the authors mentioned that a saturation of impedance change is likely due to the binding of all antibodies to the antigens. If this is the case, the impedance in 50 ug/mL antibody condition should reach plateau earlier than 100 ug/mL condition. However, figure 4 and 5 showed the opposite. 

3. The authors noted that they measured the impedance baseline (hydrogel only), but why didn't they subtract the baseline values when they did impedance analysis? 

4. In line 191, the authors mentioned the low fluctuation observed in time scan experiment indicated high accuracy and stability of the device in detecting antigens. I don't think it can demonstrate the accuracy. Accuracy is related to the deviation between values obtained from multiple independent measurements, for example, the error bars in Figure 4 and 5. In the time scan experiment, although multiple measurements were conducted, these measurements were not independent in the context of antigen since the antigen was only loaded once. 

5. The authors should explain how the device detect antigens, including details on why there is an impedance decrease following introduction of antigen. 

6. The authors used hydrogel to better capture antigen, but how and why hydrogel works was not mentioned. 

7. The authors mentioned how to prepare antibody and antigen solutions, but more importantly, the authors should also mention how they introduce antibody and antigen to the device. 

8. In line 129, figures were referenced wrong. 

Author Response

Authors response to Reviewer No. 4

Comments

The authors designed an impedance sensor to detect influenza A virus antigen. The device is responsive to antigen concentration as low as 0.5 ug/mL. The idea is novel, but data interpretation in this paper needs improvement.

Response: we appreciate the feedback by this reviewer. We have indeed addressed this point and implemented further changes to address all reviewers’ comments in the revised version of the manuscript.

The following comments need to be addressed before publication.

  1. The authors did linear regression analysis for "impedance vs antigen concentration" plots in Figure 4 and 5. However, the concentrations on the x axis are not in linear scale. Why did the author do linear regression analysis? It is not quantitative. Why the decreasing trends differ between 50 and 100 ug/mL antibody conditions?

Response: we have added the following text in section (lines 169-170 of the revised manuscript):

“As shown in figure 4 and 5 the relation between impedance and concentration is not linear. We used linear regression in the plot to highlight the presence of a decreasing trend.”

  1. In line 140, the authors mentioned that a saturation of impedance change is likely due to the binding of all antibodies to the antigens. If this is the case, the impedance in 50 ug/mL antibody condition should reach plateau earlier than 100 ug/mL condition. However, figure 4 and 5 showed the opposite.

Response: Indeed, what we meant to discuss was that we observed a significant drop in sensitivity values for both devices. Overall, the devices with higher concentration of antibodies were able to maintain higher sensitivity at the presence of increasing values of Influenza A than the ones with lower concentration of antibodies. For instance, when the concentration of influenza A reaches 50µg/ml the devices with 100µg/ml antibody are able to retain an average value of sensitivity of 13.21Ohm*ml/µg while in the devices with 50µg/ml antibody the sensitivity drops to 6.19Ohm*ml/µg, which is likely associated with the higher saturation of binding sites occupied by the antigen in the devices with 50µg/ml antibody.

To clarify this aspect we have rephrased the statement in the revised version of the manuscript (lines 173-175) as follows:

These percentages highlight the higher responsiveness of the devices with higher concentration of antibodies, up to a certain threshold attributable to a saturation of their binding sites.

  1. The authors noted that they measured the impedance baseline (hydrogel only), but why didn't they subtract the baseline values when they did impedance analysis?

Response: we thank the reviewer for this valuable comment and suggestion. The data used in the impedance analysis are consistent with the formula used to compute the sensitivity values, where the baseline impedance values are subtracted from the actual values of the measured impedance to obtain the impedance change in the numerator. The formula is included in the caption of Figure 4, and repeated here for convenience of consultation:

  1. In line 191, the authors mentioned the low fluctuation observed in time scan experiment indicated high accuracy and stability of the device in detecting antigens. I don't think it can demonstrate the accuracy. Accuracy is related to the deviation between values obtained from multiple independent measurements, for example, the error bars in Figure 4 and 5. In the time scan experiment, although multiple measurements were conducted, these measurements were not independent in the context of antigen since the antigen was only loaded once.

Response: We thank the reviewer for this important clarification. We rephrased the sentence by removing the words “accuracy” and “accurately” on this specific case (lines 226-227 of the revised manuscript).

  1. The authors should explain how the device detect antigens, including details on why there is an impedance decrease following introduction of antigen.

Response: we thank the reviewer for this observation. To address this we added the following in section 4.3 in lines 300-308 of the revised manuscript:

“The Anti-Avian Influenza A Hemagglutinin antibody is a synthetic peptide corresponding to a segment near the N terminus of the Hemagglutinin protein. The antibody specifically recognizes the cleaved subunit of Hemagglutinin but not the full-length H5. Hemagglutinin (HA) is located on the virus's surface and plays a key role in virus entry into the host cell by binding to receptors.

In electrochemical impedance spectroscopy sensors, a difference in the electrical signal is created due to the kinetic binding of antibodies and its antigens at the sensors surface. As a result, electron transfer/charge transfer resistance is produced, representing the amount of bound molecules.[18]”

and added the following reference:

[18] Magar, H.S.; Hassan, R.Y.A.; Mulchandani, A. Electrochemical Impedance Spectroscopy (EIS): Principles, Construction, and Biosensing Applications. Sensors 2021, 21, 6578. https://doi.org/10.3390/s21196578

  1. The authors used hydrogel to better capture antigen, but how and why hydrogel works was not mentioned.

Response: we thank the reviewer for this observation. To address this we added the following text in the revised version of the manuscript (lines 72-79):

“To prevent damage to the biomaterial and ensure consistent results, a hydrogel was introduced on the IDE electrodes. The hydrogel serves as a virus capturing medium, retaining the virus within the sensor's active area during measurements and providing measurement stability to the transducer. Initial experiments revealed that depositing antibodies without hydrogel, especially in small quantities at room temperatures, led to the rapid evaporation of the biomaterial from the electrode. The adoption of a hydrogel membrane over the sensor active area provides a semiwet, three-dimensional ideal environment for molecular-level biological interaction.”

  1. The authors mentioned how to prepare antibody and antigen solutions, but more importantly, the authors should also mention how they introduce antibody and antigen to the device.

Response: we used a calibrated pipette with fixed volume of 0.1 µl. We have addressed this comment by adding this detail at the end of the sub-section 2.2.3 of the “materials and methods” section (lines 107-110) as follows:

“All tests and validation experiments reported in this manuscript were carried out by using a calibrated pipette from DLAB for introducing a fixed volume of 0.1 µl of analyte solution at different concentrations to the sensor’s active area.”

  1. In line 129, figures were referenced wrong.

Response: We thank the reviewer for spotting this. We have corrected the figure reference in the text at line 164 of the revised manuscript as “Figures 4 and 5”

Round 2

Reviewer 1 Report

Comments and Suggestions for Authors

Accept in the present form.

Reviewer 3 Report

Comments and Suggestions for Authors

thank you for the revision

Reviewer 4 Report

Comments and Suggestions for Authors

The authors have addressed my comments.